# Decompression and Interlaminar Stabilization for Lumbar Spinal Stenosis: A Cohort Study and Two-Dimensional Operative Video

**DOI:** 10.3390/medicina58040516

**Published:** 2022-04-05

**Authors:** Olivia E. Gilbert, Sarah E. Lawhon, Twila L. Gaston, Jared M. Robichaux, Gabriel Claudiu Tender

**Affiliations:** Department of Neurosurgery, Louisiana State University Health Sciences Center, New Orleans, LA 70112, USA; ogilbe@lsuhsc.edu (O.E.G.); slawho@lsuhsc.edu (S.E.L.); tgast1@lsuhsc.edu (T.L.G.); jrob22@lsuhsc.edu (J.M.R.)

**Keywords:** lumbar spinal stenosis, neurogenic claudication, spine, interlaminar stabilization, lumbar laminectomy, low back pain, visual analog scale, disability index, clinical outcomes

## Abstract

*Background and Objectives*: Lumbar spinal stenosis is one of the most common causes of disability in the elderly and often necessitates surgical intervention in patients over the age of 65. Our study aimed to evaluate the clinical efficacy of interlaminar stabilization following decompressive laminectomy in patients with lumbar stenosis without instability. *Materials and Methods*: Twenty patients with lumbar stenosis underwent decompressive laminectomy and interlaminar stabilization at our academic institution. Clinical outcomes were measured using the visual analog scale (VAS) and Oswestry disability index (ODI) at the 2-month, 6-month, and 1-year postoperative visits, and these outcomes were compared to the preoperative scores. *Results:* The average VAS scores for low back pain significantly improved from 8.8 preoperatively to 4.0, 3.7, and 3.9 at 2 months, 6 months, and 1 year postoperatively, respectively (*p* < 0.001). The average VAS scores for lower extremity pain significantly improved from 9.0 preoperatively to 2.7, 2.5, and 2.5 at 2 months, 6 months, and 1 year postoperatively, respectively (*p* < 0.001). The average ODI scores significantly improved from 66.6 preoperatively to 23.8, 23.3, and 24.5 at 2 months, 6 months, and 1 year postoperatively, respectively (*p* < 0.001). There was no statistical significance for difference in VAS or ODI scores between 2 months, 6 months, and 1 year. One patient had an intraoperative durotomy that was successfully treated with local repair and lumbar drainage. Another patient had progression of stenosis and had to undergo bilateral facetectomy and fusion. *Conclusions*: Decompressive laminectomy and interlaminar stabilization in patients with spinal claudication and low back pain is a good surgical option in the absence of instability and may provide significant clinical improvement of pain and functional disability.

## 1. Introduction

Lumbar spinal stenosis is a progressive narrowing of the spinal canal, typically due to degeneration and the aging spine [1]. This mechanical compression of the neural and vascular components of the canal often results in neurogenic claudication, which greatly contributes to symptomatic patient presentation [2]. A significant cause of disability in the elderly, lumbar stenosis often necessitates spine surgery in patients over the age of 65 [3,4]. The prevalence of lumbar stenosis is known to increase with age, with individuals under 40 years of age having a 20% prevalence and individuals between 60 and 69 years of age having a 47% prevalence [5].

While age is the leading risk factor for lumbar stenosis, other risk factors include obesity, tobacco use, repetitive spinal stress with occupation, and congenital predisposition [1]. The clinical presentation can vary in severity, with some studies showing up to 9.3% of individuals being asymptomatic [6]. In the primary care setting, many patients with lumbar stenosis can be managed conservatively. Healthy lifestyle modification, medication therapy (i.e., corticosteroids, NSAIDS, opioids), epidural steroid injections, and physical therapy are all widely accepted conservative methods for initial treatment of lumbar stenosis with neurogenic claudication.

In patients with diagnosed lumbar stenosis and neurogenic claudication who fail conservative management, the option to undergo surgery is next to be considered. If the patients have no axial low back pain, a simple lumbar decompressive laminectomy is sufficient. Patients with moderate to severe stenosis due to instability typically benefit from an instrumented fusion after decompression. This, of course, is a more laborious operation and carries a higher incidence of risks, including infection, potential for pseudoarthrosis, and need for reoperation [7]. In patients with spinal claudication and low back pain, but no instability, a safe and effective alternative to fusion may be dynamic interlaminar stabilization following decompression. 

Interlaminar stabilization with the Coflex device (Surgalign, Deerfield, IL, USA) provides optimal clinical outcomes in patients diagnosed with lumbar stenosis and presenting with neurogenic claudication and axial low back pain, in the absence of frank instability. Once surgical decompression has been performed, the interlaminar device is placed between two adjacent spinous processes, with flanges on the superior and inferior aspects of the device to anchor on the superior and inferior spinous processes, respectively. It offers dynamic stabilization by permitting some degree of motion in addition to distraction of the posterior spinal elements. In general, this has been reported to have less surgical complication than standard lumbar fusion [8,9,10]. In addition, it has been briefly reported to have short- and long-term efficacy in relieving low back pain and neurogenic claudication due to lumbar spinal stenosis [8,9,10,11,12,13].

The aim of the present study was to evaluate the clinical outcomes in patients who underwent lumbar laminectomy and interlaminar stabilization for symptomatic lumbar stenosis without major instability. We include a high-quality, two-dimensional operative video to supplement our detailed description of surgical technique for a thorough understanding of this surgical option.

## 2. Materials and Methods

### 2.1. Patient Cohort

Each patient was evaluated extensively before reaching the decision to undergo decompressive laminectomy with Coflex. The inclusion criteria for patients in this study were as follows:Clinical presentation with neurogenic claudication and significant axial low back pain;Body Mass Index (BMI) of 35 kg/m^2^ or less;Evidence of lumbar stenosis on magnetic resonance imaging (MRI);Absence of spondylolisthesis, or grade 1 spondylolisthesis unchanged on flexion-extension imaging;Failure of maximal conservative management for greater than six months;Decompressive laminectomy and interlaminar stabilization at one or two lumbar levels;Completed follow up at two months, six months, and possibly one year postoperatively.

Axial low back pain in this cohort was deemed to originate from the facet joints, as most of the patients reported temporary relief with facet blocks during their conservative management. Patients with previous spine surgery or significant trauma were excluded from the study. 

A total of 20 patients between September 2014 to August 2020 fulfilled these criteria and were included in this retrospective analysis. All patients who underwent decompressive laminectomy with Coflex at our academic institution within the study timeframe were included in this study. All operative interventions were performed at a single academic institution by a single academic neurosurgeon (GCT). Most of the cohort was male (72%) and over 60 years of age (75%). The cohort BMI was uniformly under 35 per the study inclusion criteria, with average BMI of 27.6 kg/m^2^. Most operations occurred at a single lumbar level (70%), while the remainder occurred at two lumbar levels (30%). Of the single level operations, nine of them were at L4–5 (7%), with the other three being L3–4 (25%). See Appendix A for the full datasheet.

### 2.2. Surgical Technique

The operative technique is presented for the L4–5 level. The patient is placed in prone position under general anesthesia. Lateral fluoroscopy is used to identify the level of interest (L4–5) and the skin incision is centered on the two respective spinous processes. The lumbar fascia is opened with the Bovie cautery and a subperiosteal dissection is performed in order to expose the L4 and L5 laminae. The microscope can now be brought into the operative field, although some surgeons may prefer to operate under loupe magnification. The supra- and interspinous ligament is removed with the Bovie cautery and then, using the high-speed drill, a partial caudal L4 laminectomy and cranial L5 laminectomy is performed. Particular attention is paid to try to preserve as much of the L4 spinous process as possible. The yellow ligament is exposed and then removed in a piecemeal fashion with the Kerrison 2 rongeurs, from its origin underneath the cranial aspect of the L5 lamina to its insertion under the mid-L4 lamina. The midline dura mater is initially exposed. The decompression is then carried out laterally into the neural foramina, and the take-off of the spinal nerves is thoroughly decompressed. We prefer to place a piece of Gelfoam over the exposed dura for hemostasis and protection. After this, an interspinous trial is placed between the spinous processes of L4 and L5 to determine the appropriate device size (between 8 and 16 mm). The interlaminar implant (Coflex) is then inserted under fluoroscopic guidance and the two flanges on both sides are squeezed onto the L4 and L5 spinous processes, respectively. This should allow for a solid anchoring of the implant. The procedure can be repeated in a similar fashion for the adjacent level (L3–4 in this case, as the L5-S1 cannot be treated with this device). The wound is then closed in anatomical layers. The final placement of the interlaminar device(s) is confirmed by AP and lateral fluoroscopy. The surgical technique is illustrated in the operative video.

### 2.3. Evaluation Methods

To evaluate clinical efficacy, intraoperative and postoperative complications were recorded. Follow up clinic visits were completed at two months, six months, and one year. During the preoperative visit and follow up period, visual analog scale (VAS) and Oswestry disability index (ODI) scores were tabulated and recorded in the dataset for each patient. These scores were compared to determine the patients’ clinical progression. 

The VAS is a well-described and validated scale that utilizes an unmeasured, 10-cm instrument to measure pain intensity. To participate, patients indicate a point along the continuum correlating to their current level of pain. The continuum ranges from “no pain” on the left to “very severe pain” on the right. Acceptable cut points based on the distribution of VAS scores across various postsurgical patients have been recommended: no pain (0–0.4), mild pain (0.5–4.4), moderate pain (4.5–7.4), and severe pain (7.5–10.0) [14].

The ODI is a validated outcome measure commonly used in the management of persistent spinal disorders [15]. It is a Likert-scale questionnaire that broadly measures several parameters of disability, including pain intensity, personal care, lifting, walking, sitting, standing, sleeping, sexual activity, social life, and travel ability. For each section, an individual can score between 0 (least disability) to 5 (most disability). Sections scores are tabulated for an overall composite score and then translated into percentages. The percentages are stratified into five groups to describe the patient’s level of functional disability, with 0–20% indicating minimal disability, 21–40% indicating moderate disability, 41–60% indicating severe disability, 61–80% indicating crippled, and 81–100% indicating bed bound. 

### 2.4. Statistical Analysis

Data were analyzed on STATA 17.0. Paired *t*-tests were used to identify differences between preoperative and follow up VAS and ODI scores. Two-sample *t*-tests with equal variances were used to identify whether VAS or ODI were different between genders. Linear regression was used to determine the relationship between VAS or ODI scores and age or BMI.

## 3. Results

### 3.1. Clinical Outcomes

All 20 study participants were included in the statistical analysis. At the time of final follow up, no patients showed loosening of the Coflex device on flexion-extension X-ray (0.0%).

Patients’ low back pain (VAS), lower extremity pain (VAS), and overall disability (ODI) were significantly improved postoperatively. However, there was no significant improvement in pain (VAS) or disability (ODI) over time after the initial 2-month period.

The average VAS score for low back pain preoperatively was 8.8 on a 10-point scale. Postoperative scores were reduced to 4.0, 3.7, and 3.9 at 2 months, 6 months, and 1 year, respectively (*p* < 0.001). The average VAS score for lower extremity pain preoperatively was 9.0, which improved postoperatively to 2.7, 2.5, and 2.5 at 2 months, 6 months, and 1 year, respectively (*p* < 0.001) (Figure 1).

Using the ODI scale, the average preoperative patient classified as “crippled,” with a score of 66.6. Postoperatively, average ODI scores were reduced to the “moderately disabled” range, with scores of 23.8, 23.3, and 24.5 at 2 months, 6 months, and 1 year, respectively (Figure 2).

A positive correlation was found between age and preoperative lower extremity pain (VAS) (R-squared = 0.3405, *p* = 0.0087), meaning that older patients had more lower extremity pain before surgery. Postoperatively, however, there was no correlation between age and lower extremity VAS scores at 2 months, 6 months, or 1 year (*p* = 0.35, 0.41, 0.30, respectively). There were no correlations between preoperative or postoperative VAS or ODI scores and BMI or gender. All *p*-values for linear regression and two-sample *t*-tests were above a significance level of 0.05. 

### 3.2. Complications

Two patients in our cohort suffered complications (10.0%). One patient had a small durotomy during the decompression stage of the operation, resulting in a cerebrospinal fluid leak. The leak resolved after direct intraoperative repair and placement of a lumbar drain. Another patient, upon follow up, re-presented with worsening low back pain. The patient was deemed to have recurrent stenosis and ultimately underwent an instrumented lumbar fusion. His low back pain resolved after the subsequent operation and was therefore deemed attributable to the lack of initial support provided by the Coflex device. The reoperation rate in our cohort was consequently 5.0%. The complication rate in our cohort is consistent with that of the existing literature, which reports rates up to 12.1% [9]. However, data have consistently shown significantly lower complication rates with interlaminar stabilization compared to standard lumbar fusion [8,9,10]. 

### 3.3. Illustrative Case

A healthy 52-year-old male presented to the neurosurgical clinic with a 1-year history of progressive axial low back pain and bilateral neurogenic claudication. The patient has a BMI of 16.3 kg/m^2^. His neurological exam showed no focal deficits, and he had a negative straight leg raise test. The patient’s VAS scores for axial pain and lower extremity pain at presentation were 8.0 and 7.0, respectively. His ODI score at presentation was 58, classifying him as “severely disabled”. MRI of the lumbar spine showed a grade I spondylolisthesis at L3–4 with associated moderate-severe stenosis (Figure 3).

Lumbar flexion-extension films showed no overt instability on the L3–4 spondylolisthesis (Figure 4).

As part of conservative management, the patient experienced notable, but temporary, relief of his axial back pain after bilateral L3–4 medial branch blocks. Thus, the axial pain was deemed to originate from the facet joints. Maximal conservative management was exhausted for over 6 months, and thus the decision was made to move forward with surgical intervention. A decompressive laminectomy with Coflex interlaminar stabilization at L3–4 was performed without complication.

The patient’s postoperative hospital stay was undemanding as well, and he was discharged home on postoperative day 1. He subsequently completed follow up clinic appointments through 2 years with self-reported resolution of symptoms. Post-operative films display successful hardware placement without post-operative instability (Figure 5).

His postoperative VAS scores for axial low back pain were 2.0, 3.0, and 4.0 at 2 months, 6 months, and 1 year, respectively. Postoperative VAS scores for his lower extremity pain were 2.0, 2.0, and 2.0 at 2 months, 6 months, and 1 year, respectively. His ODI scores were reduced to 16, 18, and 20 at 2 months, respectively, which now classify him as “minimally disabled”. Overall, we report successful treatment of both axial low back pain and bilateral neurogenic claudication in this patient by the addition of interlaminar stabilization to a single-level decompressive laminectomy.

## 4. Discussion

Patients with lumbar stenosis typically present with neurogenic claudication and respond well to surgical decompression [2]. While radicular and claudication symptoms tend to improve reliably, improvement of back pain tends to be less predictable following laminectomy [15,16,17,18,19]. Specifically, Williams et al. found that post-operative VAS scores for back pain were found to improve an average of 1.66 at 1 year postoperatively, compared to leg pain VAS scores improving an average of 3.33 at 1 year postoperatively [17].

In addition, our data support that laminectomy with Coflex provides an effective treatment for patients with severe axial low back pain. In a study by Jones et al., 63 patients underwent decompressive laminectomy for lumbar stenosis, with back pain VAS scores assessed preoperatively, 6 weeks postoperatively, and 1 year postoperatively [20]. Prior to surgery, 31.75% (20/63) of patients had a VAS of 7–10 [20]. At 6 weeks, 15.87% (10/63) of patients reported back pain VAS scores of 7–10, and 19.05% (12/63) of patients reported back pain VAS scores 7–10 at 1 year following surgery [20]. Another study of 222 patients by Masuda et al. showed average pain scores for low back pain and leg pain to slightly worsen between 3 months and 1 year after decompression without fusion, regardless of their preoperative pain scores or disability status [21]. A retrospective analysis of 406 similar patients reported a 22% reoperation rate within 6 years mostly secondary to disease progression [22]. Similarly, a study in Finland found about 13% of patients required reoperation due to recurring symptoms [23]. Among others, these studies show a propensity of patients with severe preoperative back pain to have residual postoperative back pain in alarming rates, and therefore frequently require reoperation.

In our cohort selected specifically for severe back pain, 100% (20/20) patients presented with back pain VAS scores of ≥7.0 preoperatively, with the average being 8.8. Zero patients returned with back pain VAS of ≥7.0 at 2 months or 6 months postoperatively, with average back pain VAS scores being 4.0 and 3.7, respectively. One patient presented to the 1-year follow up appointment with a back pain VAS score of 8.0 and recurrent stenosis that was successfully treated with lumbar fusion. Despite this, the average back pain VAS score at 1 year postoperatively remained 3.9. These findings support the role and efficacy of laminectomy with Coflex in patients with severe preoperative back pain as measured by VAS scores. However, further studies including direct comparison with randomization would be needed to confirm the superiority of laminectomy with Coflex versus laminectomy alone.

In our data, the lack of change in the VAS and ODI scores at 2, 6, and 1 year suggests that the benefit from decompression plus Coflex is obtained almost immediately and sustained over time. Therefore, in patients with significant back pain without overt instability on dynamic imaging, decompressive laminectomy with interlaminar stabilization remains a viable option to address claudication symptoms in combination with axial back pain, where decompression alone may not be effective. Interlaminar stabilization with the Coflex device offers an intermediate option between the simple decompression and the instrumented fusion. 

The addition of the Coflex device after laminectomy typically adds minimal operative time, with one study citing operative times of 141.91 ± 47.88 min for laminectomy with Coflex vs. 106.81 ± 41.30 min for laminectomy alone [24]. In addition, the procedure provides an alternative to an instrumented fusion, which may be too extensive an operation, carrying with it the risk of misplaced hardware and adjacent segment degeneration due to the change in the flexible dynamics of the spine. Davis et al. found that laminectomy with Coflex had significantly shorter operative time, blood loss, and length of stay compared to lumbar fusion [25]. Moreover, if a fusion is deemed necessary at a later time, the Coflex device can be easily removed, and the placement of an interbody cage and posterior hardware is not impeded in any way. 

The Coflex device differs from previous interspinous implants (e.g., X-stop, Diam) in that it is not rigid, and the connecting part of the device is located in between the remaining portion of the laminae following decompression, rather than the spinous processes, thus being closer to the axis of rotation. These differences represent a proposed mechanism of why the failure rate, though high with the interspinous devices, has been minimal with the interlaminar dynamic devices. Another reason for low failure rates observed with interlaminar stabilization may be the low level of activity in the elderly population. The average age of our patients was 63 years, and their preoperative activities were severely restricted by their lumbar symptoms, not uncommonly being wheelchair bound. After the operation, these patients progressively resume their daily activities, but these activities rarely amount to more than walking to the store or working in the garden. If the same symptoms occurred in younger individuals, an instrumented fusion may become the better option. Finally, success seen in our utilization of interlaminar stabilization with decompressive laminectomy can be related to the usage of this type of stabilization in patients with a body mass index of 35 kg/m^2^ or less. This was done to avoid the overwhelming stress that a large upper body weight would inflict on the interlaminar device. In patients with BMI over 35 kg/m^2^, an instrumented fusion was offered if deemed necessary, especially in patients presenting with focal neurologic deficits or bowel or bladder incontinence. Otherwise, bariatric surgery with close clinical follow up was offered to decrease the risk of surgical complications that is well documented in this patient population [26]. 

There are limitations to this study. The main limitation is the short follow up period after surgery. Therefore, long-term outcomes on the time scale of five or more years in our specific cohort are not validated. However, there are a few existing studies that support its long-term efficacy in patients with lumbar stenosis, with successful follow up measured at 8 years and 12 years [11,13]. Regarding our participants, the extensive inclusion criteria for this study inherently limited the number of patients who were eligible for analysis. Secondly, our participants represent a purposive cohort sample from a single academic institution. Although their clinical profiles and outcomes are consistent with current literature, interpretation of results is limited. Future prospective studies might further characterize the exact patient population which would benefit from interlaminar stabilization with surgical decompression for lumbar spinal stenosis. 

## 5. Conclusions

Decompressive laminectomy and interlaminar stabilization in patients with spinal claudication and axial low back pain is a good surgical option in the absence of instability. Overall, VAS and ODI scores were significantly improved at 2 months, 6 months, and 1 year follow up in our study cohort. This technique may be particularly useful in elderly patients with limited physical activity. 

## Figures and Tables

**Figure 1 medicina-58-00516-f001:**
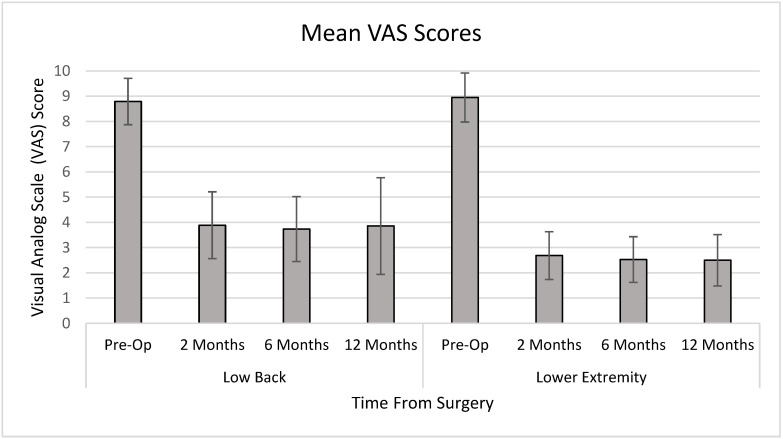
Mean VAS scores for low back pain and lower extremity pain over time.

**Figure 2 medicina-58-00516-f002:**
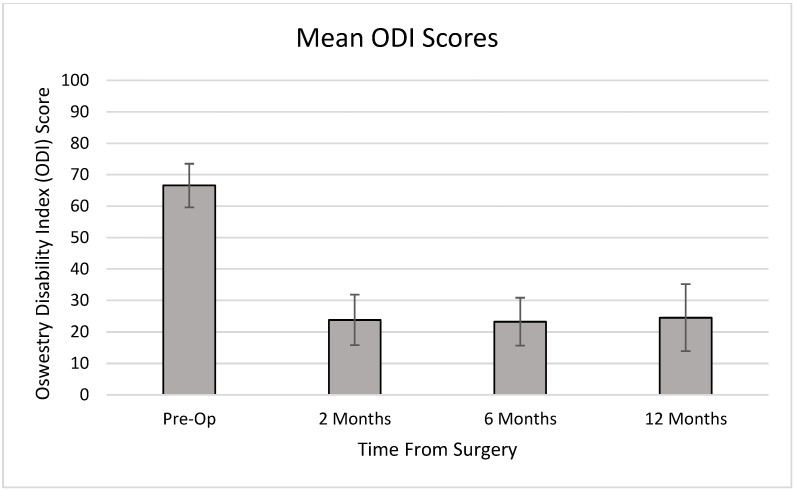
Mean ODI Scores over time.

**Figure 3 medicina-58-00516-f003:**
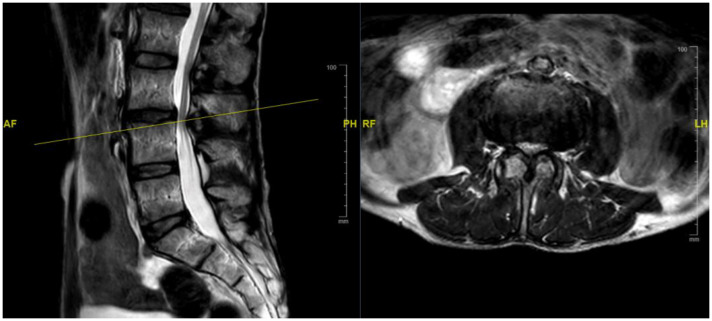
MRI lumbar spine, sagittal (**left**) and axial (**right**) cuts, showing grade I spondylolisthesis and associated moderate-severe stenosis at L3–4.

**Figure 4 medicina-58-00516-f004:**
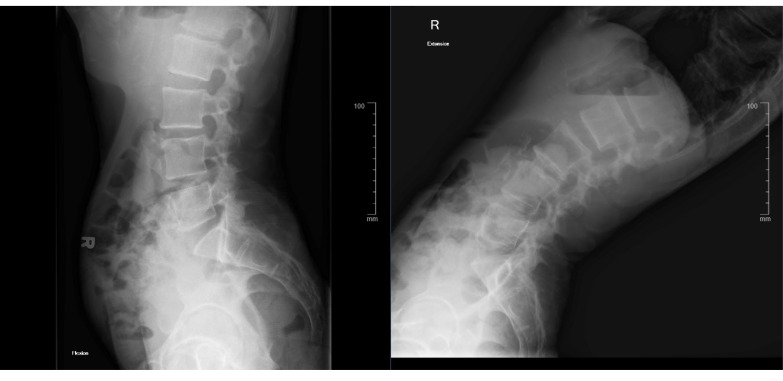
Flexion-extension X-ray showing no overt instability.

**Figure 5 medicina-58-00516-f005:**
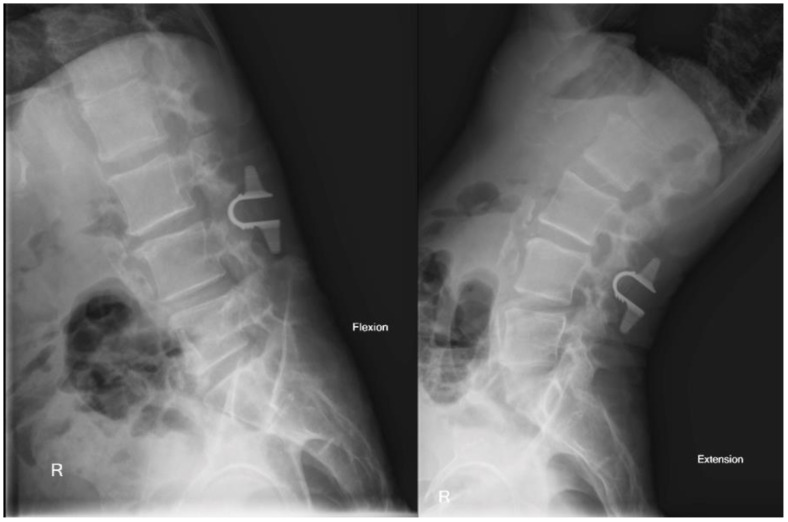
Postoperative flexion and extension X-ray at 1 year follow up showing effective hardware placement without iatrogenic instability.

## Data Availability

The data presented in this study are available in Appendix A.

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
