# Peer review of "Decompression and Interlaminar Stabilization for Lumbar Spinal Stenosis: A Cohort Study and Two-Dimensional Operative Video"

_medicina, 2022, doi:10.3390/medicina58040516_

Round 1
Reviewer 1 Report
The control in the literature is only one paper. This is not insufficient and the reviewer cannot agree. Authors need to show at least the data from a several papers.
Others;
1) No pain, mild pain, moderate pain, severe pain: Authors should show these data by a Table in the text.
2) Line 252: Is “50-60%” correct?
Author Response
The control in the literature is only one paper. This is not insufficient and the reviewer cannot agree. Authors need to show at least the data from a several papers.
In addition to the 12 citations we included in the Introduction and Discussion sections regarding this specific comparison (previous #s 2; 7-13; 15-18), we have added 5 more citations to support our statements (new #s 18-19; 21-23). See Lines 239, 248-256.
Others;
1) No pain, mild pain, moderate pain, severe pain: Authors should show these data by a Table in the text.
The pain scores are reported in Figure 1 and are detailed more in Supplementary Table 1. The authors did not use these categorical breakdowns for statistical analysis; the authors’ point in the Methods (Lines 138-141) was merely stated for completeness.
2) Line 252: Is “50-60%” correct?
Yes, but it was deleted in Line 256.
Reviewer 2 Report
The manuscript now presents an adequate data analysis of back pain which significantly improved the strength of the message. Illustrative cases also helps in supporting the thesis discussed.
Author Response
Thank you!
Reviewer 3 Report
The correction process is visible, the article has been expanded with additional descriptions in the results and discussion parts.
This is the basis for accepting the article to the publication in the present form.
Author Response
Thank you!
This manuscript is a resubmission of an earlier submission. The following is a list of the peer review reports and author responses from that submission.
Round 1
Reviewer 1 Report
The authors propose an interesting and controverse topic, but to me there are some flaws in the setting of the study. A control group of patients should be considered since, considering the simple stenosis characterizing the patient sample, a stand-alone decompression (without interlaminar device) should be the first-choice treatment. Outcome differences between two homogeneous groups in this sense would give strength to the paper. Another concern regards the use of Coflex in cases without instability and with the purpose of treating axial back pain. The authors are encouraged to support this indication with strong argumentation, considering the well-known overcharge of the anterior column, with accertained consequences on disc units, by placing a device between the spinous processes. How the authors distinguish when the axial back pain would benefit for a Coflex? If the axial pain, as in many of the cases, is a discal pain interspinous devices are not recommended due to the aforementioned biomechanical unbalance. Do the authors test if the pain trigger regards the facet joints (for instance with infiltrations) and if so, they consider the placement of the Coflex? Generally speaking, how they discern when the axial back pain is not disc-related and eligible for interspinous device with benefits? The patient sample, lacking a control group, is relatively limited and it is not homogeneous considering that single and double level stenosis are mixed in the same pool. This could represent a significant bias since the analysis is based on VAS and ODI parameters. I would suggest focusing the study only on single level stenosis (the majority of patients), hopefully including more patients, to compare with a proportional group of single level stenosis treated only with a spinal canal decompression.
Author Response
We sincerely thank the reviewers for the time and effort put behind their suggestions and have addressed each one carefully. We hope our revisions are satisfactory and would be happy to resolve any additional comments, if necessary.
Reviewer 1:
The authors propose an interesting and controverse topic, but to me there are some flaws in the setting of the study. A control group of patients should be considered since, considering the simple stenosis characterizing the patient sample, a stand-alone decompression (without interlaminar device) should be the first-choice treatment. Outcome differences between two homogeneous groups in this sense would give strength to the paper.
Unfortunately, we do not believe it feasible to have a control group for this particular study. Historically, patients with spinal claudication AND axial facetogenic pain have not done well with a simple decompression, hence the indication for Coflex. This is supported with current literature (Lines 51-58, 66-69, 199-205, 209-212) and reflected in our extensive inclusion criteria (Lines 78-94). We agree with the reviewer that not having a control group poses certain implications to our results and have recognized the purposive cohort sampling scheme as a limitation to the study (Lines 251-253).
Another concern regards the use of Coflex in cases without instability and with the purpose of treating axial back pain. The authors are encouraged to support this indication with strong argumentation, considering the well-known overcharge of the anterior column, with accertained consequences on disc units, by placing a device between the spinous processes. How the authors distinguish when the axial back pain would benefit for a Coflex? If the axial pain, as in many of the cases, is a discal pain interspinous devices are not recommended due to the aforementioned biomechanical unbalance. Do the authors test if the pain trigger regards the facet joints (for instance with infiltrations) and if so, they consider the placement of the Coflex? Generally speaking, how they discern when the axial back pain is not disc-related and eligible for interspinous device with benefits?
Indeed, the patients we included in this study underwent significant work-up before surgical selection (Lines 77-78). The inclusion criteria are thoroughly detailed in the Materials and Methods (Lines 78-94). We have added a sentence to confirm the axial low back pain was indeed deemed to originate at the facet joints, as most of these patients underwent facet blocks with temporary relief (Lines 91-93). The MRIs showed relatively preserved disc height at the treated levels.
The patient sample, lacking a control group, is relatively limited and it is not homogeneous considering that single and double level stenosis are mixed in the same pool. This could represent a significant bias since the analysis is based on VAS and ODI parameters. I would suggest focusing the study only on single level stenosis (the majority of patients), hopefully including more patients, to compare with a proportional group of single level stenosis treated only with a spinal canal decompression.
Overall, we agree with the reviewer that our cohort is relatively small and limited without a comparable control group (Lines 249-253). However, the inclusion criteria were carefully devised by the authors to include patients with one- and two-level stenosis (Lines 87-88) based off the precedents of current literature and general recommendations for the use of the Coflex device. Therefore, we believe it is satisfactory to analyze them as a single group considering their similar clinical pathology and outcomes.
Reviewer 2 Report
The subject of interest is one of the methods of correcting spinal stenosis.
To the authors:
It's a very small population.
Improve your charts, label the chart axes, enter the standard deviation range.
Please explain why each VAS result for the lower back and lower extremities, as well as ODI scores after the intervention, did not change during follow-up at 2, 6, and 12 months.
Author Response
We sincerely thank the reviewers for the time and effort put behind their suggestions and have addressed each one carefully. We hope our revisions are satisfactory and would be happy to resolve any additional comments, if necessary.
Reviewer 2:
The subject of interest is one of the methods of correcting spinal stenosis.
To the authors:
It's a very small population.
We agree with the reviewer regarding our study’s limited sample size. However, we were very selective with the inclusion criteria (Lines 78-94), and thus the number of patients who underwent Coflex was inevitably small. We have expanded upon this limitation in the Discussion to further clarify the tempered nature of our results (Lines 249-253). Despite this limitation, this study offers guidance on the use of Coflex in a specific population with lumbar stenosis and axial facetogenic pain, upon which future studies can build.
Improve your charts, label the chart axes, enter the standard deviation range.
We have recreated Figures 1 and 2 to include standard deviation ranges and labels for the chart axes (Lines 169, 176).
Please explain why each VAS result for the lower back and lower extremities, as well as ODI scores after the intervention, did not change during follow-up at 2, 6, and 12 months.
We thank the reviewer for this suggestion and have expanded the Discussion section to address these results (Lines 205-209). The lack of change in the VAS and ODI scores at 2, 6, and 12 months suggests that the benefit from decompression plus Coflex is obtained almost immediately (by 2 months). The fact that VAS and ODI remain stable over time suggests that the construct continues to provide support for at least 1 year.
Reviewer 3 Report
Thank you very much for giving me to review your work titled “Decompression and Interlaminar Stabilization for Lumbar 2 Spinal Stenosis: A Cohort Study and 2-Dimensional Operative 3 Video”. This time, the reviewer cannot judge this article is sufficient for publication in Medicina or not.
The authors should make a major revision.
- There is no control study. Authors should compare your findings of this study with the results of DECOMPRESSIIN ALONE and FUSION SURGERY much more.
- Complications should be clearly separated and compared between those related to decompression procedures and those related to implants.
- As authors mentions, one year follow-up period is too short. At least, authors should describe the segmental mobility on flexion-extension of roentgenogram and loosening of device before surgery and the final follow-up.
- How about the original number of patients treated by Coflex®? There is no mention about that.
Author Response
We sincerely thank the reviewers for the time and effort put behind their suggestions and have addressed each one carefully. We hope our revisions are satisfactory and would be happy to resolve any additional comments, if necessary.
Reviewer 3:
Thank you very much for giving me to review your work titled “Decompression and Interlaminar Stabilization for Lumbar 2 Spinal Stenosis: A Cohort Study and 2-Dimensional Operative 3 Video”. This time, the reviewer cannot judge this article is sufficient for publication in Medicina or not.
The authors should make a major revision.
1. There is no control study. Authors should compare your findings of this study with the results of DECOMPRESSIIN ALONE and FUSION SURGERY much more.
As discussed above, we unfortunately were not able to have control group. Historically, patients with spinal claudication AND axial facetogenic pain have not done well with a simple decompression, hence the indication for Coflex. Similarly, patients undergoing a fusion have more extensive pathology and/or instability, hence they could not be compared to the Coflex group.
2. Complications should be clearly separated and compared between those related to decompression procedures and those related to implants.
We have clarified these aspects in the Complications subsection. The dural tear was related to decompression (Line 187). The recurrence of low back pain and need for a subsequent fusion was likely related to the Coflex device not being able to provide sufficient support (Lines 192-193).
3. As authors mentions, one year follow-up period is too short. At least, authors should describe the segmental mobility on flexion-extension of roentgenogram and loosening of device before surgery and the final follow-up.
Thank you for this suggestion. We have added that none of the patients showed device loosening on flexion-extension X-rays at their final follow-up (Lines 157-159).
4. How about the original number of patients treated by Coflex®? There is no mention about that.
These were ALL the patients who underwent Coflex between 2014 and 2020 (Lines 95-96). None of the Coflex patients were excluded. We have reenforced this detail in the Materials and Methods (Lines 96-98) as well as the Results (Lines 157).
Round 2
Reviewer 1 Report
The manuscript adequately improved, but, since the authors report an indication for Coflex based on pain generation from facet joints, I recommend the integration of a teaching case with images (preferably both X-rays and MRI) before and after treatment with the purpose of emphasizing their concepts especially with imaging at 12 months.
Figure 1 and 2 both show a double graph. Please erase of them in each figure.
Author Response

(The authors gave the same response as above.)

Reviewer 3 Report
In order to improve the quality of manuscript, the reviewer still requests major revision.
- The authors cite Reference 15 for comparison. But why don't the authors compare the data with the same analysis as the citation? If authors do it, it should be easier to understand the benefit of authors' operative procedure.
- There should be many reports that decompression alone improves back pain. The reviewer thinks the author's opinion should be quoted as well.
Author Response
Reviewer 3: In order to improve the quality of manuscript, the reviewer still requests major revision.
1. The authors cite Reference 15 for comparison. But why don't the authors compare the data with the same analysis as the citation? If authors do it, it should be easier to understand the benefit of authors' operative procedure.
We have extensively revised our Discussion to better portray our data when compared to some existing data on laminectomy alone (Lines 236-241; 245-263).
2. There should be many reports that decompression alone improves back pain. The reviewer thinks the author's opinion should be quoted as well.
We have revised a few phrases in the Discussion to detract the unintended conclusion that decompression alone does not ever improve axial back pain (Lines 237-239; 243-245). While discussing a few new references, we simply advocate that there is an unfortunately high number of patients (50-60%) with axial low back pain who undergo decompressive laminectomy and inevitably return with recurring pain (Lines 237-252). We emphasize our study’s data to state that the addition of interlaminar stabilization to laminectomy allows for longer-lasting decreases in VAS scores, and that the axial pain relief is almost immediate for many patients (Lines 253-266).